# Evaluation of the Florida Newborn Screening Program Education Campaign

**DOI:** 10.3390/ijns11010020

**Published:** 2025-03-10

**Authors:** Mirine Richey, Cynthia B. Wilson, Minna Jia, Travis Galbraith

**Affiliations:** 1Center for Prevention and Early Intervention Policy, Florida State University, 1339 E Lafayette St., Tallahassee, FL 32301, USA; m.richey@fsu.edu; 2The Florida Center for Prevention Research, Florida State University, 2200 Old St. Augustine Rd., Tallahassee, FL 32301, USA; 3FSU Survey Foundry, #6147 University Center, Florida State University, 296 Champions Way, Tallahassee, FL 32306, USA; minna.jia@fsu.edu; 4Milken Institute School of Public Health, George Washington University, Washington, DC 20052, USA; tgalbraith@gwmail.gwu.edu

**Keywords:** newborn screening, focus groups, survey, perceptions, parental awareness

## Abstract

Florida’s Newborn Screening Program campaign aims to increase the awareness and participation of birthing facilities, providers, and parents. This evaluation aimed to determine the effectiveness and reach of the Newborn Screening Program (NBS) Statewide Educational Campaign to pregnant women through surveys and focus groups. The online survey, conducted throughout Florida in English, Spanish, and Haitian Creole, evaluated the reach and effectiveness of educational materials such as paid advertisements and brochures. The surveys also served to recruit participants for in-person focus groups throughout the state. The findings showed that 85.3% of the mothers had discussions with health professionals about the screening program, while others did not hear about it from health professionals. More than 50% of the respondents learned about the program through health facilities, with additional exposure from media platforms such as television, radio, and friends. This study shows the need for increased outreach of the campaign and better communication and education from medical professionals to increase awareness.

## 1. Introduction

Newborn screening refers to a framework including diagnostic process performed on a newborn 24–48 h after birth to identify potential health conditions, enabling timely identification, early referral intervention, and a care plan for treatment [1]. The screening is performed in order to identify infants at risk for specific medical conditions that may threaten the child’s health, development, and well-being. While screening is a first step towards appropriate intervention, a system ensuring the timely coordination of care remains just as crucial [2]. Overseeing the system and framework of screening and care coordination in the state of Florida is the Florida Department of Health (DOH). The disorders targeted by the screening, and subsequent processes, are in state statute [3] and are generally those that, without intervention, would cause significant morbidity, mortality, or intellectual disability [4], impacting families. The Florida Department of Health (DOH) oversees the implementation and monitoring of Florida’s Newborn Screening Program (NBS), which can be performed in the birth facility, physician’s office, or at home for planned homebirths. As of 2023, Florida’s newborn screening program comprises three primary tests: blood spot screening tests to check the baby’s blood for selected conditions, pulse oximetry screening to check for any heart abnormalities, and a hearing screening test to detect hearing levels.

All babies are offered testing since even babies who appear asymptomatic may have some complications and benefit from early referral. Within the Florida primary tests, there are screenings for 35 core conditions, following the Recommended Uniform Screening Panel (RUSP) in the United States [5], and 22 secondary conditions [6]. The core conditions include hearing loss, endocrine disorders, and metabolic disorders. Metabolic disorders make up the bulk of diseases being screened for, with fatty acid oxidation disorders and amino acid disorders among the conditions identified. In most instances, symptoms of inborn metabolic disorders may appear in early infancy, although some may become more apparent in late childhood.

The hearing test and pulse oximetry, to screen for critical congenital heart defects (CCHDs), are non-invasive painless tests carried out at the facility prior to discharge, and then results are received through the provider once they are ready to ensure prompt referral and care. The program is funded through a joint effort, where hospitals and birthing facilities pay USD 15 for every live birth. Medicaid and private insurance companies are billed for the tests, and the state covers the costs for uninsured families [6].

The program has a significant impact in reducing the risk of genetic, metabolic, and other congenital conditions that may affect the infant after birth. The Florida NBS established a benchmark goal that less than 1% of all specimens received at the laboratory be unsatisfactory in terms of the submission being incomplete or having an unsatisfactory blood spot specimen for testing. However, the current unsatisfactory rate in Florida ranges between 1.0% and 1.5%. Another benchmark goal set by the NBS is that all specimens be received at the laboratory no later than 3 days after collection, although not all specimens are received within this timeframe. These performance benchmarks are monitored and reported to the submitting facilities monthly, with quarterly grades posted on the NBS’s public-facing website (floridanewbornscreening.com).

The screening program tests over 225,000 samples annually, and of these, over 700 babies are identified as having a condition that will benefit from early detection and treatment. Many newborn conditions, such as phenylketonuria, cystic fibrosis, and sickle cell disease, are managed through the newborn screening program in collaboration with Children’s Medical Services. Some conditions, like X-linked adrenoleukodystrophy (X-ALD), require follow-up testing and a timely response to optimize outcomes [4,7]. The scope of newborn screening continues to expand, with additional conditions regularly added to the screening panel by state statute, including the most recent addition in 2024, of Cytomegalovirus (CMV). Parents can opt out of newborn screening, which should include a written refusal that will be included in the medical record [6]. Research on parent willingness, attitudes, and beliefs show that some parents would refuse to participate in newborn bloodspot screening to avoid pain for the baby due to their previous experiences with healthcare services, the thought that the tests are unnecessary, religious reasons, and not trusting the government with the child’s DNA [8,9]. It is crucial to consider factors concerning the parent and child data and how they are handled and how this will impact the decision to take part in the blood screening. Previous studies investigated parental psychosocial impacts on newborn screening results, including true positives and inconclusive results [8].

Healthcare providers, while mandated to offer newborn screening to all parents, may influence parent decision-making or parent perceptions [10]. Because of potential healthcare provider influence on newborn screening success rates, previous studies demonstrated that various educational messaging be available from different sources including public information campaigns aimed at parents [11]. Florida’s NBS provides healthcare provider education, and a public campaign aimed at pregnant women and families to address gaps in the Florida screening rates.

Florida NBS also reports on hospital, obstetrician, and midwife screening performance measures based on samples received in a quality- and time-sensitive manner to the Bureau of Public Health Laboratories in Jacksonville. The performance measures provide the public with information to make an informed decision on the provider and location. The aims of this study are to evaluate (a) how well the Florida NBS education campaign reached the target parent audience, (b) how and if new and expectant mothers in Florida interacted with the NBS educational campaign, and (c) how the campaign influenced the attitudes, knowledge, and intent regarding the screening of their newborn. The campaign aims to increase awareness of the importance of newborn screening and the conditions covered in Florida.

## 2. Materials and Methods

A mixed-method approach was used for this evaluation, including an online survey and focus groups. The survey included only quantitative (closed-ended) questions. Focus groups aimed to provide a more comprehensive picture of what possible influence the Newborn Screening Program (NBS) educational campaign may have had on the target population. The survey and focus group questions were designed to complement one another. That is, the survey question asked the “what” questions, while the focus groups questions asked the “how”, “when”, and “why” questions to elicit a better understanding from participants. In addition to collecting demographic information, survey and focus group questions determined whether participants saw the NBS educational campaign, the medium from which they saw the campaign (different information sources), knowledge, and facts about newborn screenings, and how the education materials influenced their decision to screen their newborns or plan to screen their newborns within the recommended timeframe. The target population for the survey and focus groups included pregnant women currently residing in Florida and women who had given birth in Florida during the campaign evaluation period of 2023.

### 2.1. Recruitment Procedures

The Florida Center for Prevention Research (FCPR) developed a recruitment flyer for interested participants to access and complete the online survey. The flyer was posted in English, Spanish, and Haitian Creole and included an overview of the study, an invitation to participate, and a description of incentives. Participants could access the survey either through a scannable QR code on the survey or a shortened website URL. The Center for Prevention and Early Intervention Policy (CPEIP) collaborated with Florida home visiting programs, Florida Women, Infants and Children’s (WIC) programs, Florida Healthy Start programs, and other maternal and child health (MCH) outreach in Florida to disseminate recruitment materials to the target population. CPEIP requested the groups post the flyer in print, in all available languages, at their offices and on their social media. Programs within the focus group target counties assisted in passing the information to the women they serve within their agencies. Other recruitment efforts included communicating with the doctors’ offices where the Florida Department of Health (DOH) had previously placed the brochures, posters, and other NBS educational materials to assess their willingness to be part of the recruitment.

Additionally, sponsored Facebook ad posts focused on reaching the target audience were posted to reach a broader demographic. Although this did yield a significant number of responses, it should be noted that there was an unexpectedly high number of responses that, upon further investigation, were determined to be invalid responses. The FSU Survey Foundry (FSUSF) found responses suspected of being “bot” activity when it was detected that multiple responses came from same IP address and were duplicative responses. Based on numerous articles addressing “bot” activity in survey research [12,13,14], strategies were implored to increase the validity of responses, including a reCAPTCHA verification option in the Qualtrics system. The initial recruitment of online flyers with the gift card promotion is suspected to have been a driver of the “bot” activity. Therefore, those survey responses were eliminated from the dataset. The target completion rate for the online survey was 600 respondents. This sample size was based on the number of responses needed to provide enough data for meaningful analysis and to draw reliable conclusions. To encourage a higher response rate, an incentive of a USD 5 Amazon gift card was offered to the first 600 verified participants.

### 2.2. Focus Group Recruitment

Participants in the online survey were invited to participate in a 45 to 60 min in-person focus group. Participants were informed that they would receive a USD 25 Amazon gift card for participating in the focus group. At the end of the online survey, using a separate link that could not be matched to their survey responses, interested participants were prompted to share their names, emails, phone numbers, and the best time to reach them. Upon receiving the information of potential participants, the FCPR contacted them via email and text message to share more details about the focus group and confirm their willingness to participate. Once they confirmed, information on each focus group was sent to participants two weeks before the date. Several reminders were sent to participants, including one week prior and one day prior, via email and text message.

The FCPR and CPEIP aimed to recruit participants from diverse backgrounds to the focus groups, including women from the state’s rural, urban, and suburban areas. Focus groups were conducted from seven target regions throughout Florida (Leon, Jackson, Dade, Duval, Orange, Palm Beach, and Hillsborough counties). Focus groups were hosted in collaborating MCH program offices in each county. Due to the low attendance rate at the in-person focus groups and the number of requests received for virtual focus groups, the FCPR attempted to hold additional virtual focus groups for those who could not attend in person. However, although many women registered, no one participated, and this was determined by the research team to be an attempt to obtain the incentive gift card without participating. For example, several emails were received asking for the gift card although no documentation of participation existed for those persons. The target number of focus groups was six, with an average of ten people per group. A total of seven in-person focus groups were held. The actual number and characteristics of focus group participants are discussed further in Section 3.

### 2.3. Data Collection

Survey Instrument. In a collaborative effort that began in March 2023, the DOH Children’s Medical Services (CMS) team and all three FSU research centers collaborated and designed the survey questionnaire (Appendix A). This collaboration aimed to create a tool that was both comprehensive and sensitive to the nuances of public health messaging. The final instrument consisted of 21 multiple-choice questions, carefully formulated to gauge the public’s awareness of the campaign and its effectiveness. The survey was developed using an online survey platform (Qualtrics), which could easily be accessed with a shortened web address or scannable QR code. The survey included English, Spanish, and Haitian Creole language options, significantly broadening accessibility and inclusivity for Florida’s diverse populations. Questions focused on assessing participants’ awareness of the Florida Newborn Screening Program, their engagement with the Educational Campaign, and their knowledge of newborn screening procedures and benefits. The survey was designed to be clear, concise, and user-friendly, ensuring respondents could provide accurate and meaningful responses. Data security and respondent privacy were prioritized throughout the survey design. No personally identifiable information was collected from respondents. Before data collection, a Human Subjects Determination form was submitted to the Florida State University Institutional Review Board (IRB).

It was determined that this study protocol qualified the study for exemption; therefore, approval to proceed was given by the FSU IRB.

Survey Administration. The FSU Survey Foundry (FSUSF) administered the online survey between 1 November 2023 and 9 December 2023. The survey was strategically deployed during high engagement times to maximize respondent participation and ensure robust data collection. Since a complete sampling frame was unavailable, the survey employed a convenience sampling strategy, as discussed in the Recruitment section above. This approach facilitated rapid data collection from a readily accessible population subset.

Focus Group Instrument. The FCPR developed and approved focus group questions for DOH, the funding agency, to ensure the evaluation aligned with the department’s objectives (Appendix B). These questions guided the focus group discussions at each session, although the participants were free to provide additional feedback.

Focus Group Implementation. Focus groups were between 45 and 60 min in length. Sign-in sheets were collected at each session, including the date of the session, participant’s name, and signature. Participants were asked to complete a Focus Group Participant Info form to gather information needed to have complete contact information for sending incentives. Sessions began by thanking participants for coming, introducing our team, and introducing the purpose of the focus group. Participants were then shown a series of NBS campaign materials, including print materials, a screenshot of the NBS website, and a brochure. Next, they were shown each NBS education campaign video (30 and 60 s versions) and the radio advertisement. Participants were given a Public Education Campaign Material Review form to record whether they recalled seeing any of the items shown (available through floridanewbornscreening.com). All focus groups were audio-recorded and transcribed. No personally identifiable information was recorded on the audio recording or transcriptions, so responses would not be connected to individual respondents. Instead, participants were labeled as “participant” on the transcript. While the surveys were in English, Spanish, and Haitian Creole, focus groups were held in English-only due to staff limitations.

### 2.4. Data Analysis

Surveys. Survey data were analyzed using descriptive statistics and cross-tabulations, to answer the research questions outlined in this Evaluation Plan. This included providing demographic data of respondents, assessing the level of awareness and knowledge among the respondents, and comparing responses according to age, ethnicity, and education.

Focus Groups: Focus group recordings were transcribed using Descript. Transcriptions were then verified by manually confirming the transcription against the audio recording. A thematic approach was employed to summarize each focus group’s key themes. To contextualize themes identified in the data, axial coding was used to identify connections among major code categories from the data across all focus groups. Two separate researchers conducted independent coding to ensure inter-rater reliability.

## 3. Results

### 3.1. Survey Respondent Characteristics

From a total of 1406 responses, we validated and analyzed 628 responses. The remaining responses were excluded after rigorous screening identified instances of “bot” activity, fake accounts, and other anomalies previously mentioned, which were removed to maintain data quality and accuracy. Studies detailing the challenges of unreliable survey feedback due to increased “bot” activity suggest that having 45% of responses determined to be valid is a feature to be expected [15,16]. Detailed demographic characteristics that are vital for understanding the campaign’s reach and impact were summarized.

Survey respondents (65%) were between 20 and 30 years old, followed by 33% in the 30–40 age group. Only 1% of respondents were over 40 years or under 20 years. Notably, mothers under the age of 30 made up over half of the respondents, a demographic particularly relevant to the objectives of the Newborn Screening Program Campaign.

In terms of racial and ethnic composition, 57.3% of participants identify as White, while 33.3% identify as Black or African American. Smaller proportions include 3.5% American Indian or Native Alaskan, 1.9% Asian, and 2.4% Native Hawaiian or Pacific Islander.

Educational attainment among respondents shows the largest proportion (37.6%) having completed a 4-year college degree. This is followed by 23.4% who have completed 2 years of college and 17.7% with some college education. Fewer respondents reported having a professional degree or technical school education (9.9%), a high school diploma (8.4%), or less than a high school education (2.2%).

The breakdown of responses by language is as follows: 19 in Haitian Creole, 3 in Spanish, and the remainder in English.

### 3.2. Summary of Key Survey Themes

The following summarizes key themes identified from the survey, followed by relevant tables that outline the survey data. Appendix C includes a complete set of all data tables. Relevant tables to the overall results and discussion are included in the main text. Notably, 85.2% of the mothers surveyed reported discussing the Newborn Screening Program with healthcare professionals at various stages of their maternity journey, highlighting effective communication channels (see Table 1). While 69.3% were given the option to opt out of the screening, 17% were not given this option, and 12.9% were unaware of it, indicating areas where patient education could be improved (see Table 2).

### 3.3. Awareness of the Campaign

Medical professionals (doctor’s office, clinic, or hospital) informed more than half (58%) of the respondents about the Newborn Screening Program, underscoring the critical role that hospitals and clinics play in disseminating information. Traditional media outlets such as TV (15%) and radio (9.6%), as well as personal networks including family and friends (9.6%), also contributed to spreading awareness, indicating a well-rounded outreach strategy (see Table 3).

Overall, 75.3% of respondents recalled seeing the NBS advertisements, with variations by age suggesting the campaign reached its intended audience. Of those who recalled seeing the advertisements, 97.6% reported that the information was easy to understand, reflecting the campaign’s success in engaging its audience.

### 3.4. Importance of the Newborn Screening

The overwhelming majority (91.5%) recognized the importance of Newborn Screening, responding that it was either “very important” or “important” underlining the program’s perceived value among new and expecting mothers and reinforcing the public health message’s penetration and acceptance. Only a minimal fraction (1.7%) did not consider the screening important, suggesting widespread support for the initiative. In general, the higher the education, the higher the awareness of the importance of newborn screening.

Sources of Information. The survey also explored trust in information sources, with results indicating a strong preference for digital platforms among mothers. Social media and online resources emerged as the most trusted channels, pointing to the importance of maintaining robust, clear, and scientifically accurate online content.

### 3.5. Focus Group Participants

A total of 164 individuals located in regions convenient to the location of the focus groups indicated their interest in participating in a focus group on the online survey. All 164 individuals were contacted via email and text with details about the focus group, a reminder of the incentive, and a request to register for an upcoming focus group in their area. The registration process intended to gather information to ensure they met the inclusion criteria and to gather additional contact information. Multiple messages were sent to these individuals to encourage them to register. A total of 35 individuals registered for a focus group. Once they registered, additional emails and text messages were sent with specific details for the focus group that they registered for, with a request to confirm their attendance. Only 12 individuals confirmed attendance. In the end, a total of 13 individuals participated in the focus groups across all regions, although less than a typical focus group (8–10 people per group) target number of participants for the project, participation in research from new mothers has been historically challenging due to new parent responsibilities and lack of time [17,18], and the responses received were still enlightening and provided helpful qualitative data.

Summary of Key Focus Group Themes. Appendix D summarizes key themes from each focus group including a summary table of responses, including some quotations, of each question asked during the focus groups. The following provides a brief synthesis of key themes summarizing all the focus groups cumulatively.

Focus Group Participants. Participant Demographics: Focus groups included both first-time mothers and those with multiple children, providing diverse perspectives.

### 3.6. General Awareness of Newborn Screening

Mixed Awareness: Participants showed varying levels of awareness about newborn screening. Most focus group participants (75%) had heard about newborn screening, while a few others learned about it during the focus groups. One common theme was that many mothers did not have a name for the Newborn Screening Program, but after explaining the screening procedures, they recalled this happening after childbirth.

Sources of Information: Information was sporadically provided by medical professionals. Some participants did not receive detailed discussions on newborn screening during previous pregnancies.

Notable quote:

“*I think we’re just overwhelmed with so much information when we’re pregnant, especially for the first time, that we see a lot of it. We take a lot of it in, but at least I know I didn’t really have much register.*”

### 3.7. Effectiveness of Educational Campaign Materials

Recognition: There was mixed recognition of campaign materials like logos, posters, brochures, and radio/television advertisements. The sources with the highest rate of recognition included Facebook, posters, and the NBS brochure. The source with the lowest rate of recognition was television advertisements, with only a few reporting hearing the radio ads.

Participants indicated that television and radio ads were less impactful due to consumption habits of women in their 20s and 30s. They noted that they typically stream their television media without ads, or if they are listening to the radio, they change the channel as soon as an ad comes on.

Participants communicated limited exposure to campaign materials, a total of 8 of the 13 participants (62%) reported seeing at least some of the NBS campaign materials. This underscores the importance of utilizing various dissemination channels for future campaign materials to increase the chances that materials will reach the target audience.

Design and Content: Participants gave positive feedback on the use of bright colors and engaging visuals (especially pictures of babies). Participants preferred straightforward messages about the importance of newborn screenings. When shown a screenshot of the NBS webpage, respondents provided positive feedback about the design and ease of navigation.

Preferred Media: Participants preferred social media, posters, and brochures with bright colors and engaging visuals (especially pictures of babies). Videos were appreciated but less frequently encountered. Participants indicated the videos were well done, but most stated they mostly watch television and listen to music on streaming services, so they do not encounter television or radio advertisements.

Suggestions for Improvement: Participants suggested greater visibility and distribution of materials in hospitals, clinics, and Healthy Start offices, and inclusion of detailed, easy-to-understand information in brochures. One notable suggestion was to place video advertisements on televisions in hospital rooms or in provider’s waiting rooms. They suggested more focus should be given to target the intended audience through social media rather than traditional television and radio ads. They also recommended including more diversity in images.

### 3.8. Hospital Experiences with Newborn Screening

Varied Experiences: Hospitals provided varied information about newborn screening. Some participants received detailed explanations and experienced reassurance from the information they received, while others received minimal information during the actual screening or with their discharge papers. One participant recalled receiving login information to check results after discharge and appreciated this being communicated by the hospital. A general sentiment was that newborn screening seemed to be a common, routine experience during childbirth, so they did not question it. In fact, not one focus group participant questioned the necessity of newborn screening.

Communication with Medical Professionals: Direct discussions with doctors and nurses were valuable but inconsistent across healthcare facilities, with some participants stating the nurses explained the procedure while it was being performed, while others had very little information given to them. Sources of information included obstetricians, nurses performing tests, childbirth classes, and hospital tours. A common theme from mothers was that more information may have been shared with them, but because there is so much information coming at them so quickly, they may have not retained the information.

The reported experiences from focus groups were slightly different from the surveys. Over 85% of survey respondents recalled a medical professional communicating with them about newborn screening, either before, during, or after childbirth. In comparison, 9 of 12 (75%) focus group participants who responded to this question indicated that a medical professional had spoken with them about newborn screening. Since the focus groups had a limited sample size, the results are not generalizable, and variations from surveys with a larger sample size are to be expected.

Notable quotes:

“*It just seemed routine and expected, the medical professionals didn’t say much before doing it.*”

“*Everybody was amazing and…explained what was going on…. She was like ‘she passes!’*”

“*It made me glad that they were doing it. Especially as a first-time parent, you don’t know what to expect…so having them come in and explain what was happening…reassures us.*”

## 4. Discussion

Participants expressed high levels of trust in medical professionals for health information but noted a desire for more detailed and accessible explanations. While previous evaluations of the Florida campaign have not been undertaken, previous studies cite needs for improvement of parent engagement in the entire process including promotion, consent, awareness and timely referral and follow-up care [19,20,21]. Several participants, particularly those from rural areas, indicated the need to seek care outside of their county, in larger cities. This highlights the need for targeted rural outreach, a strategy shown to be effective in previous NBS campaigns that combined healthcare provider engagement with social media platforms like Facebook and Instagram to reach medically underserved areas [22,23]

While friends’ advice was valued, participants considered it secondary to professional guidance, particularly when the friend lacked personal experience with childbirth. Trust in digital sources, such as Google and social media platforms, varied widely. Social media emerged as a preferred and frequently used medium for information, with many participants citing it as their first exposure to the NBS. However, concerns about the inconsistency of information from Google searches led participants to cross-reference multiple websites, such as Baby Center, to verify accuracy. Despite ongoing efforts to improve the reliability of online health information [24], the burden of determining accuracy remains on the user. The strategy of using point-of-care marketing and health education messaging available to patients in portals, waiting rooms, and exam rooms on free-standing computer screens, television, and tablets, is a common practice in the U.S. For example, community-based home visiting models in the U.S. are required to use as standardized curriculum. Curriculum such as the FSU *Partners for a Healthy Baby* allows for the delivery of materials, including information on newborn screening, directly via text messaging, email, or printed handout delivery [25] and is used throughout the U.S. and Florida. Other point-of-care programs are the Expecting Health platform (expectinghealth.org) which may be recommended directly to the patient from the provider, or the new StrongFLMoms.com that has launched in cooperation with the Florida Department of Health, both of which are available on any mobile device.

Considerations for digital health communication and integration into patient electronic healthcare records (EHR) could bridge communication gaps directly from the healthcare provider to improve the patient point of contact services and reduce the need for the patient to seek additional outside information.

Participants provided actionable suggestions for enhancing future NBS campaigns, including increased use of social media and streaming services like Spotify for public service announcements, as these were seen as more effective than traditional media such as cable TV or radio. They also recommended earlier integration of focus group feedback in developing educational materials to ensure relevance to the target audience. For example, several participants emphasized the need for materials featuring more diversity and cultural representation.

Focus group discussions further highlighted the importance of diversity in communication and dissemination channels. Materials with the highest recognition rates included Facebook posts, posters, and brochures, while video and audio advertisements were less effective in reaching the target audience. While traditional television was explored in this study, the use of waiting-room patient education video broadcast was not part of the NBS campaign. Future recommendations could explore the use of patient video education and engagement in the obstetrical exam and waiting room [26] as an additional option for exploring collaborations with existing. Participants recommended that future campaigns prioritize a mix of formats to ensure broader reach, and patient point-of-care broadcast services seen on televisions and computer displays could fill a gap in reach.

Despite recognizing hospitals and physicians as primary sources of education about NBS, participants emphasized the need for more proactive and transparent communication during prenatal and postnatal care. They expressed a desire for detailed information about the purpose and scope of NBS, as well as timely access to screening results. One participant noted, “*I need to know what exactly they’re screening for, and not just being told we’re gonna* [sic] *prick your heel… and not really know why*.” This feedback underscores the need for more thorough patient education and engagement before and during hospital stays. Finally, participants identified gaps in healthcare provider communication, suggesting that future evaluations may want to include Florida’s provider dashboard data and grading scale to provide a more robust illustration of the gaps among healthcare and birth facilities. While this falls outside the scope of the current evaluation, such measures could enhance trust and satisfaction in future initiatives.

## 5. Conclusions

The collaboration to better understand the level of awareness of the Florida Newborn Screening Program (NBS) and the effectiveness of the Statewide Florida NBS Educational Campaign through quantitative and qualitative measures found that, overall, there is a general awareness of the Florida NBS, although many respondents reported a lack of detailed knowledge about the purpose and scope of newborn screening.

### Limitations

The challenges encountered with online surveys, such as the need for the elimination of duplicate survey responses and monitor for “bot” activity were not part of the initial considerations, although corrected for early on, responses had already been received. The promotion of incentivized survey taking in research should be monitored closely, or a strategy for a secondary incentive collection method considered. Additionally, the study design introduces the potential for recall bias in that participants may not have accurately remembered all the details of their newborn screening experience, affecting the true reflection of campaign or provider efficacy.

While the majority of survey and focus group respondents reported having discussions with healthcare professionals about the NBS at various stages of their maternity journey, misunderstandings about the process remain. Survey respondents and focus group participants were consistent in how they learned about NBS, with their top two ways of learning being the hospital (during prenatal tours or childbirth classes) or their obstetrician/clinic.

In terms of reaching the primary target audience of pregnant women and new mothers with the NBS educational campaign, findings were mixed between the survey and focus groups; however, there is a general awareness of the NBS.

The high recall rate of advertisements and their clarity underscore the campaign’s success in engaging and educating its audience. The overwhelming recognition of the importance of newborn screening among respondents highlights the campaign’s impact on public perception and acceptance.

The mixed methods approach enabled an evaluation that included both breadth and depth. However, it should be noted that while focus group data offer valuable insights into the collective perspectives and experiences of participants, it is essential to recognize the inherent limitations. The findings are often context-specific, influenced by group dynamics, and may not be generalizable to broader populations.

Additionally, the subjective nature of qualitative data analysis can introduce potential biases. Despite these limitations, the focus groups provided a powerful tool for exploring nuanced issues surrounding the experiences with NBS and the reach of the educational campaign that complemented the online survey.

Based on these findings, we concluded that the Florida NBS Educational Campaign demonstrated moderate effectiveness in increasing awareness and understanding of the importance of newborn screening, contributing to the health and well-being of infants across the state. Future efforts should focus on enhancing communication strategies to better reach the target audience by addressing the barriers identified during the evaluation, including broader dissemination of educational materials to ensure the intended target audience is exposed to information about newborn screening. Continued collaboration and rigorous evaluation will be essential in sustaining and improving the impact of the Florida NBS educational campaign.

## Figures and Tables

**Table 1 IJNS-11-00020-t001:** Did any medical professionals talk to you about the purpose and benefits of the Florida Newborn Screening Program before, during, or after pregnancy?

Response	Frequency	Percent
Yes	535	85.2
No	61	9.7
I’m not sure	31	4.9
Total	627	99.8

**Table 2 IJNS-11-00020-t002:** Were you given the option to refuse newborn screening?

Response	Frequency	Percent
Yes	435	69.3
No	107	17.0
I’m not sure	81	12.9
Total	623	99.2

**Table 3 IJNS-11-00020-t003:** How did you learn about the Newborn Screening Program?

Response	Frequency	Percent
TV	94	15.0
Radio Station	60	9.6
Doctor’s office or Clinic	160	25.5
Hospital	204	32.5
Family and friends	60	9.6
Other (Please Specify)	12	1.9
Total	590	94.1

## Data Availability

The original contributions presented in this study are included in the article. Further inquiries can be directed to the corresponding author.

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
