# Peer review of "Evaluation of the Florida Newborn Screening Program Education Campaign"

_2409-515X, 2025, doi:10.3390/ijns11010020_

Round 1

Reviewer 1 Report

Comments and Suggestions for Authors

Thank you to the authors for undertaking this study. The methods were well implemented and presented. Comments below reflect areas needing to be addressed from this reviewer's perspective. 

INTRODUCTION

From this reviewer’s perspective, citations in the introduction are extremely sparse and needs significant work to incorporate other important references (i.e. see where only citation 5 is provided, etc).

[See similar comment recommended for Discussion. Important to acknowledge what has already been done, and perhaps why this work was needed and is still needed to be implemented.]

First sentence Line 30-32

“Newborn screening refers to the testing carried out on infants immediately after birth 30

to ensure they are safe and healthy through the early detection and treatment of health

issues.”

From this reviewer’s perspective, the sentence as worded needs modification, particularly for readers who may be less familiar with the NBS system. NBS is a “process” for which “testing” is one part (as Sue Berry often points out). Plus, “to ensure they are safe and healthy” is not precise enough as there are NBS conditions for which the child may be identified yet never be fully “healthy.” Certainly the NBS goal is to have “benefit”, yet that does not necessarily mean “healthy” as an outcome. Please consider rewording to address both aspects noted.

Line 38:

“it comprises…”

Please identify early on that this work is limited to FL. Please also be specific what “it” refers to (FL NBS? RUSP?).

Line 59-60:

“benchmark goal that less than 1% of all specimens received at the laboratory be unsatisfactory”

Please clarify what “be unsatisfactory” means

Line 68-71:

Words are missing in this sentence

--MATERIALS AND METHODS:

Line 135

“…upon further investigation, were determined to be invalid responses.”

Please give a few brief examples (i.e., missing required data?

Line 159-160

“…although many women registered, no one participated, and

this was determined to be an attempt to fraud FSU for the incentive money.”

While stated in the following sentence more recruitment info is discussed in the Results section, from this reviewer’s perspective mentioning “fraud” warrants an immediate brief clarificatory sentence in this recruitment section as well.(incl clarifying the word used by authors: “determined” : by whom? The authors? FSU Counsel, the state?)

Line 166

“meticulously designed the survey”

From this reviewer’s perspective, such subjective terms as “meticulously” from authors’ considering their own efforts should be deleted.

Line 210

Focus groups were only held in English due to staff limitations.

RESULTS

Line 225-228

“From a total of 1,406 responses, we validated and analyzed responses. The remaining responses were excluded after rigorous screening identified instances of “bot” activity, fake accounts, and other anomalies, which were removed to maintain data quality and accuracy.”

From this reviewer’s perspective, this merits a statement as to whether this is consistent with other research (incidence, and citation if available).

Line 292-293

“Although this number was significantly less than the target number of participants…”

From this reviewer’s perspective, this merits a statement as to whether this is consistent with other research (incidence, and citation if available).

Line 340-341

“One notable suggestion was to place video advertisements on televisions in hospital rooms or in provider’s waiting rooms.”

From this reviewer’s perspective, this merits a comment in later section --with citations-- to highlight this suggestion has been mentioned by others many times over the years yet unfortunately is not often implemented

Line 369-370

Please similarly italicize first quote as had for others

Line 409-410

Please be consistent and similarly italicize quotes for readability

EXPERIENCES IN HOSPITALS

From this reviewer’s perspective, this is confusing to provide a Header (4) for a category so different than all the other headers (i.e., Methods, Results, Discussion)

CONCLUSION

From this reviewer’s perspective, this section merits a separate subsection with LIMITATIONS.

Please also address the “fraud” issue here and perhaps lessons learned to inform future research (and other researchers)

From this reviewer’s perspective, along with previously raising that citations in the introduction are extremely sparse and needs significant work to incorporate other important references (i.e. see where only citation 5 is provided, etc) this comment is applicable for Discussion. Important to acknowledge what has already been done, and perhaps why this work was needed and is still needed to be implemented.

CITATIONS

As noted, insufficient from this review's perspective yet relatively easy to remediate.

Please also note that this reviewer did NOT check or review any statistical details

Author Response

Good afternoon,  thank you for the detailed questions and edit suggestions.  I appreciate the opportunity to improve the material to be relevant to the readers. Please see the attached response and resubmission as well. 

Sincerely, Mirine Richey

Reviewer 2 Report

Comments and Suggestions for Authors

The Florida Newborn Screening (NBS) Program reports the results of a survey to assess the impact of an educational campaign. The manuscript is well-organized and informative. The manuscript did not define what goals the program has for NBS education or the educational campaign. This would have helped the reader evaluate the results of over 80% of parents being aware of NBS. Is the goal 100%? Also was there any relationship between parents who decline NBS and educaton? Conducting surveys in English, Spanish, and Haitian Creole is commendable, as it ensures inclusivity and access for diverse populations within Florida. This approach strengthens the study’s applicability and potential for generalizability across different linguistic and cultural groups. The combination of surveys and focus groups adds depth to the analysis, allowing both quantitative and qualitative insights into maternal awareness and perception of NBS. It would have been interesting if the survey and/or focus group had distinguished between prenatal, birth, and postnatal since a useful goal of NBS education may be to ensure prospective parents are aware and informed before birth. In conclusion, the study provides valuable insights into the reach and effectiveness of Florida’s NBS educational campaign, highlighting both successes and areas needing improvement. 

Author Response

Dear reviewer, 

Thank you so much for your input and reflection on our project. I appreciate some of the point you made, and I have made incorporated some of these thoughts into our 2nd draft.  I appreciate the opportunity to strengthen the paper and use this guidance for future research aims.

Comment: This would have helped the reader evaluate the results of over 80% of parents being aware of NBS. Is the goal 100%?

Answer: This is a reasonable question to gage the impacts, however, our initial project with the funder did not include benchmarks, but rather an overall impression and impact of the campaign materials. I did mention in the (added) limitation section on the small sample size and potential for recall bias.

Comment: Also was there any relationship between parents who decline NBS and educaton?

Answer: We did not analyze refusal or those who commented they did not recall if they refused with education.

Sincerely,

Mirine Richey

Round 2

Reviewer 1 Report

Comments and Suggestions for Authors

Thank you to the authors for addressing many of the reviewer comments to benefit readers. From this reviewer's perspective, further consideration of some additional current resources would help readers. Therefore, this reviewer suggests making it a bit clearer  a few  by adding a few more relatively recent citations in the context of NBS innovative Education in the Prenatal Setting. For example,  Expecting Health; etc.

Lastly, as before, statistical analysis and/or review of the quantifiable data was not undertaken by this reviewer.   

Author Response

Thank you I added future considerations incorporating some of the trends to technology, EHR, and parent engagement to the Discussions section. Lines 416-428. I hope this amplifies the opportunities for  new engagement ideas of parents for the readers.